



# Ice-shelf ocean boundary layer dynamics from large-eddy simulations

Carolyn Branecky Begeman[1], Xylar Asay-Davis[1], and Luke Van Roekel[1]

[1]Los Alamos National Laboratory, P.O. Box 1663, Los Alamos, New Mexico, USA 87545

**Correspondence:** Carolyn Begeman (cbegeman@lanl.gov)

**Abstract.** Small scale, turbulent flow below ice shelves is regionally isolated and difficult to measure and simulate. Yet these small scale processes, which regulate heat transfer between the ocean and ice shelves, can affect sea-level rise by altering the ability of Antarctic ice shelves to "buttress" ice flux to the ocean. In this study, we improve our understanding of turbulence below ice shelves by means of large-eddy simulations at sub-meter resolution, capturing boundary layer mixing at scales
intermediate between laboratory experiments or direct numerical simulations and regional or global ocean circulation models. Our simulations feature the development of an ice-shelf ocean boundary layer through dynamic ice melting in a regime with low thermal driving, low ice-shelf basal slope, and strong shear driven by the geostrophic flow. We present a preliminary assessment of existing ice-shelf basal melt parameterizations adopted in single component or coupled ice-sheet and ocean models on the basis of a small parameter study. While the parameterized linear relationship between ice-shelf melt rate and
far-field ocean temperature appears to be robust, we point out a little-considered relationship between ice-shelf basal slope and melting worthy of further study.

## 1 Introduction

The largest source of uncertainty in future sea level rise is the potential loss of ice from the Antarctic Ice Sheet (IPCC, 2014). The rate of grounded ice loss is highly sensitive to the melting of ice shelves, which drain over 80% of Antarctica's grounded ice
(Reese et al., 2018; Rignot et al., 2013). The Ice-shelf Ocean Boundary Layer (IOBL) controls ice-shelf melting by regulating oceanic heat and salt fluxes to the ice shelf base. Thus, accurate predictions of ice-shelf melting depend on representing the turbulent dynamics of the IOBL. This representation is also critical for evaluating the sensitivity of the coupled land ice-ocean system to changes in ocean conditions. Of particular concern is the sensitivity of ice-shelf melting to increasing seawater temperature at depth, a trend observed along a wide swath of the West Antarctic coastline and a potential trigger for West
Antarctic Ice Sheet collapse (Purkey et al., 2018; Ruan et al., 2021; Schmidtko et al., 2014; Wåhlin et al., 2021).

One indication that ocean models do not capture turbulent dynamics in the IOBL is that the simulated thermal driving, the difference between ocean temperature and the local freezing point, and consequently the simulated melt rate differ significantly between ocean models and as resolution is varied within a model (Gwyther et al., 2020). Furthermore, ocean models predict ice-shelf melting using parameterizations that neglect the effects of the lateral buoyancy gradient of the IOBL, and often of the
vertical buoyancy gradient, on the efficiency of vertical mixing near the boundary. This model deficiency likely biases turbulent



fluxes through the IOBL and ocean cavity circulation, which is driven in large part by the buoyant flow of water freshened by ice-shelf melting, an overturning circulation also known as the "ice pump" (Webber et al., 2018). A new parameterization of ice-shelf melting that accounts for the dynamics of IOBL turbulence is needed to achieve a more physically based, accurate coupling of ice sheets and oceans in climate models (Dinniman et al., 2016; Edwards et al., 2021; Gwyther et al., 2020;

Naughten et al., 2018).

IOBLs present unique conditions in the global ocean, involving a stabilizing flux from phase change (ice-melting) and a boundary layer that is positively buoyant against a sloping boundary. There is a rich literature on stably-stratified boundary layers (typically under constant stabilizing flux boundary conditions), but the dependence of heat, salt and momentum fluxes on stratification remains a difficult problem, especially for strongly stratified regimes (Zonta and Soldati, 2018). The IOBL

has most frequently been explored at small scales through laboratory experiments and direct numerical simulations, with some recent numerical studies addressing intermediate scales (Middleton et al., 2021; Mondal et al., 2019; Vreugdenhil and Taylor, 2019). However, this body of work has not yet matured to setting a new standard for ice-shelf melt parameterization in ocean models. Today, the most commonly used ice-shelf melt parameterization still derives from sea-ice conditions (i.e., in the absence of a slope) with some parameters tuned for ice-shelf settings (Holland and Jenkins, 1999; Jenkins et al., 2010;

McPhee et al., 1987). A knowledge gap exists in bridging IOBL dynamics across scales and characterizing the structure of buoyant plumes. Recently, this has been addressed through non-turbulence-resolving 2-d or 2.5-d models (Cheng et al., 2020; Jenkins, 2016, 2021). Notably, Jenkins (2021) found that vertical mixing in IOBL settings was represented poorly by the commonly-used K-Profile Parameterization (KPP) in comparison to a higher-order turbulence closure scheme, suggesting that the application of KPP in ocean models may be inappropriate in ice-shelf cavities.

In this study, we model turbulent heat, salt, and momentum fluxes through the IOBL using Large-Eddy Simulation (LES). Whereas the ocean models typically used to model sub-ice-shelf circulation lack both the resolution and appropriate parameterizations needed to capture the relevant turbulent scales for boundary layer dynamics, LES captures the dominant energy-containing scales of turbulence and represents smaller, unresolved scales with varying degrees of complexity. An effective parameterization of ice-shelf melting is likely to rest on an understanding of how turbulent mixing in the IOBL depends on

stratification and shear forcing. We vary far-field ocean temperature and ice-shelf slope between model runs and characterize turbulent fluxes and ice-shelf melt rates. In this study we focus on the high-shear, low thermal driving regime. The target of previous LES studies has been on low-shear settings (Middleton et al., 2021; Vreugdenhil and Taylor, 2019). Section 2 describes the LES model, its turbulence closure scheme, and the set-up of our simulations. Section 3 presents the results of our simulations, with a focus on comparisons across the sampled range of thermal driving and slope. The discussion is given in

two parts: Section 4.1 contextualizes the features of our simulated IOBL turbulence and discusses limitations of this study, and Section 4.2 focuses on how this study informs parameterizations of ice-shelf melting and IOBL turbulence. We provide some closing thoughts in Section 5.





## 2 Methods

### 2.1 Overview of the LES model

The PArallelized Large eddy simulation Model (PALM) was developed at the Institute of Meteorology and Climatology at Leibniz Universitat Hannover, Germany (Raasch and Schröter, 2001) for simulation of atmospheric and ocean boundary layers. For this application to sub-ice-shelf settings, we developed a new version based on PALM v5.0 (Maronga et al., 2015) with added features including a boundary flux scheme for the ice-ocean interface, rotating the gravity and Coriolis vectors for sloped domains, and a different turbulence closure scheme. Here, we provide a brief overview of PALM with a focus on our additions.

For more details we refer readers to Maronga et al. (2015).

The governing equations of PALM are the following:

Momentum conservation
$$\frac{\partial u_i}{\partial t} = -\frac{\partial u_i u_j}{\partial x_i} - \varepsilon_{ijk} f_j u_k + \varepsilon_{i3j} f_3 u_{g,j}$$
$$+ g_i \frac{\rho - \rho_a}{\rho_0} - \frac{1}{\rho_0} \frac{\partial \pi^*}{\partial x_i}$$
$$- \frac{\partial}{\partial x_j} \left( \overline{u_i'' u_j''} - \frac{1}{3} \overline{u_i'' u_i''} \delta_{ij} \right) \tag{1}$$

Volume conservation
$$\frac{\partial u_j}{\partial x_j} = 0 \tag{2}$$

Heat conservation
$$\frac{\partial \theta}{\partial t} = -\frac{\partial u_j \theta}{\partial x_j} - \frac{\partial}{\partial x_j} \left( \overline{u_i'' \theta''} \right) + \Phi_\theta \tag{3}$$

Salt conservation
$$\frac{\partial S}{\partial t} = -\frac{\partial u_j S}{\partial x_j} - \frac{\partial}{\partial x_j} \left( \overline{u_i'' S''} \right) + \Phi_S \tag{4}$$

The momentum terms on the right hand side of Equation 1 are, in order: advection, Coriolis forcing, imposed geostrophic
flow, buoyancy forcing, a correction for divergence in the flow using the pressure perturbation $\pi^*$ (imposing incompressibility), and sub-grid scale momentum flux. All prognostic variables are considered filtered at the grid scale. This is typically denoted by the overbar, which we omit except for the sub-grid turbulent flux terms to emphasize that we only represent averaged effects with a turbulence closure scheme. Double primes denote sub-grid fluctuations.

We represent a sloping ice base by rotating the gravity ($\boldsymbol{g}$) and Coriolis ($\boldsymbol{f}$) vectors while keeping the domain a rectangular
prism as in Vreugdenhil and Taylor (2019). Specifically,

$$\boldsymbol{g} = g_z [\sin\alpha \sin\beta, \sin\alpha \cos\beta, \cos\alpha] \tag{5}$$

$$\boldsymbol{f} = 2\Omega [\sin\phi \sin\alpha \sin\beta, \cos\phi \sin\alpha \cos\beta, \sin\phi \cos\alpha] \tag{6}$$

where $g_z$ is the magnitude of gravitational acceleration in the geopotential z-direction, $\beta$ is the angle the up-slope direction makes with north, and $\alpha$ is the slope angle from horizontal. For the rotated Coriolis vector $\boldsymbol{f}$, $\Omega$ is the rotation rate and $\phi$ is the
latitude. Unless otherwise specified, quantities are oriented with the simulated grid.





The buoyancy term in Equation 1 combines the contributions of along-slope pressure gradient due to the slope of the ice shelf in hydrostatic equilibrium and buoyancy due to changes in density relative to the ambient density of the water column $\rho_a$. $\rho_a$ varies along slope and with depth from the ice interface as a function of the hydrostatic pressure:

$$\rho_a(x,y,z) = f_{EOS}\left(\theta^i(z), S^i(z), p(x,y,z)\right) \tag{7}$$

where the superscript $i$ denotes the initial state. The reference density $\rho_0$ is evaluated in the center of the x-y plane ($\rho_0(z) = \rho_a(x_{mid}, y_{mid}, z)$). We neglect hydrostatic pressure gradients along slope, except through $\rho_a$ in the buoyancy term. For the maximum slope simulated in this study, the maximum hydrostatic pressure gradient is on the order of $100\,\mathrm{Pa\,m^{-1}}$. This simplification has the advantage of avoiding pressure discontinuities across periodic boundary.

The terms on the right-hand side of the heat and salt conservation equations (Equations 3 and 4) are advection by the resolved
flow, turbulent transport by the sub-grid scale fluctuations, and the source and sink terms $\Phi_{\theta,S}$. The source and sink terms are zero in our simulations because we treat heat and salt fluxes due to melting at the boundaries through the sub-grid vertical flux term, as discussed in Section 2.2.

### 2.2 Turbulence closure

The turbulence closure scheme is Anisotropic Minimum Dissipation (AMD; Rozema et al., 2015) employing the extension
to scalars introduced by Abkar et al. (2016). We validated our implementation against the stable atmospheric boundary layer test case published in Abkar and Moin (2017). Typical sub-grid scale momentum and scalar diffusivities in our sub-ice-shelf simulations of stratified turbulence range from $10^{-5}$ to $10^{-4}\,\mathrm{m^2\,s^{-1}}$ in the upper two-thirds of the domain where damping is not applied (the damping methodology is discussed in Section 2.3).

Sources and sinks of momentum, heat and salt due to interactions between the flow and the ice base are all parameterized as
sub-grid fluxes at the boundary. The resolved vertical fluxes at the top layer of the model go to zero as $w$ goes to zero according to the no-flux boundary condition, and the sub-grid fluxes are determined using the scheme described below in place of the AMD scheme at the top layer. PALM is vertically discretized such that the ice boundary ($z = 0$) is located at the edge of the top cell where the vertical velocity component resides and scalars and horizontal velocity components are located at mid-depth of the top cell ($z = -\frac{1}{2}\Delta_z$). We denote the interface location with subscript $b$ and the middle of the first cell below the interface
with subscript $\frac{1}{2}$.

Subgrid momentum fluxes are parameterized according to law of the wall following a linear stability function for stabilizing buoyancy forcing as in Vreugdenhil and Taylor (2019):

$$\overline{w''u_i''}_b = u_* \frac{\kappa\left[u_i\left(z_{\frac{1}{2}}\right) - u_{i,b}\right]}{\ln\left(z_{\frac{1}{2}}/z_0\right) - \Psi_m\left(\zeta_{\frac{1}{2}}\right)} \tag{8}$$

for the horizontal velocity components (i.e., $i = 1, 2$). $\zeta_k = z_k/L_O$ is the depth from the boundary scaled by the Monin-
Obukhov length, $u_*$ is the friction velocity, and $z_0$ is the roughness length. The horizontal velocity at the boundary $u_{i,b}$ is always 0. The Monin-Obukhov length $L_O$, computed following McPhee et al. (1987), is a function of both $u_*$ and the melt





rate. The stability function $\Psi_m$ is linear with the scaled depth:

$$\Psi_m(\zeta) = 1 + \beta_m \zeta \tag{9}$$

This parameterization assumes that Coriolis forces are not locally dominant in the momentum balance.

Similarly, the sub-grid fluxes of scalars are parameterized by

$$\overline{w''\theta''}_b = \Gamma_\theta u_* \left( \theta_{\frac{1}{2}} - \theta_b \right), \, \overline{w''S''}_b = \Gamma_S u_* \left( S_{\frac{1}{2}} - S_b \right) \tag{10}$$

where the exchange coefficients are defined following McPhee et al. (1987):

$$\Gamma_\theta^{-1} = \left( \Gamma_{\theta,turb} + \Gamma_{\theta,mol} \right)^{-1}, \, \Gamma_S^{-1} = \left( \Gamma_{S,turb} + \Gamma_{S,mol} \right)^{-1} \tag{11}$$

The turbulent flux component follows a shape function consistent with the parameterization of momentum fluxes in Equation
125     8

$$\Gamma_{\theta,turb} = \kappa^{-1} \left[ \ln \left( z_{\frac{1}{2}}/z_0 \right) - \Psi_\theta \left( \zeta_{\frac{1}{2}} \right) \right], \, \Gamma_{S,turb} = \kappa^{-1} \left[ \ln \left( z_{\frac{1}{2}}/z_0 \right) - \Psi_S \left( \zeta_{\frac{1}{2}} \right) \right] \tag{12}$$

with analogous stability functions to momentum, Equation 9,

$$\Psi_\theta(\zeta) = 1 + \beta_\theta \zeta, \, \Psi_S(\zeta) = 1 + \beta_S \zeta \tag{13}$$

The coefficients of these stability functions are chosen as in Zhou et al. (2017) and are given in Table S1. The molecular flux
components are constant in our simulations following McPhee et al. (1987) and are also given in Table S1. If a portion of the
ice face experiences freezing, $\Gamma_\theta$ and $\Gamma_S$ are set to a higher value indicating destabilizing fluxes ($\Gamma_f$, Table S1).

Taking into account the role of meltwater advection, Equation 10 is replaced by the following "virtual" freshwater flux form
in our implementation (Asay-Davis et al., 2016; Jenkins et al., 2001):

$$\overline{w''\theta''}_b = -c_p u_* \left[ \Gamma_\theta - \Gamma_S \frac{S_b - S_{\frac{1}{2}}}{S_b} \right] \left( \theta_b - \theta_{\frac{1}{2}} \right) \tag{14}$$

$$\overline{w''S''}_b = -u_* \left[ \Gamma_S - \Gamma_S \frac{S_b - S_{\frac{1}{2}}}{S_b} \right] \left( S_b - S_{\frac{1}{2}} \right) \tag{15}$$

The temperature and salinity at the ice-ocean interface, $\theta_b$ and $S_b$, are unknown. Three equations are used to solve for these
quantities and the melt rate $m$, the so-called three-equation parameterization:

$$\rho c_p \overline{w''\theta''}_b = -\rho_w m L \tag{16}$$

$$\rho \overline{w''S''}_b = -\rho_w m S_b \tag{17}$$

$$\theta_b = \theta_f(p, S_b) \tag{18}$$

These equations specify that heat and salt are conserved at the ice-ocean interface (Equations 16 and 17) and the interface
temperature is fixed at the local freezing temperature $\theta_f$ (Equation 18). Equation 16 assumes that the conductive heat flux into





ice is negligible. $\rho_w$ denotes the density of freshwater. The freezing point is calculated using the polynomial function from Jackett et al. (2006).

The PALM implementation applies the fluxes $\overline{w''u_i''}_b$, $\overline{w''\theta''}_b$, $\overline{w''S''}_b$ at the center of the first grid cell from the boundary without interpolation (i.e., $\overline{w''X''}_b = \overline{w''X''}_{\frac{1}{2}}$). It is noted by the PALM developers that this error was found to be small, but we have not confirmed this for the sub-ice case. This error would be small if the first $\sim 10\,\mathrm{cm}$ were largely a constant flux layer, as hypothesized by McPhee (1983).

## 2.3    Simulation set-up

The domain is a $64\,\mathrm{m}^3$ cube with horizontal resolution of $0.5\,\mathrm{m}$ and vertical resolution of $0.25\,\mathrm{m}$. The ice-ocean interface is located on the top boundary of the domain. Large-scale horizontal pressure gradients drive the mean flow, which is geostrophic in the far-field where buoyancy does not modify the flow. We choose a pressure of $800\,\mathrm{dbar}$ at the top of the domain, an intermediate choice given that the depth of ice shelf bottoms ranges an order of magnitude (roughly $-2000\,\mathrm{m}$ to $-200\,\mathrm{m}$). The potentially dynamically relevant differences between conducting these simulations at surface pressure and $800\,\mathrm{dbar}$ is that the

first derivative of the freezing temperature with respect to salinity is 20% smaller and the first derivative of density with respect to salinity is about 2 times larger.

     Boundary conditions for velocity, temperature and salinity are periodic at the side boundaries, Dirichlet at the bottom boundary, and von Neumann at the top boundary. The bottom third of the domain is a sponge layer (Rayleigh damping) within which velocity, temperature, and salinity are relaxed toward their assigned values at the bottom of the domain (Klemp and Lilly,

1978; Maronga et al., 2015). The sponge layer results in negligible vertical fluxes of heat, salt, or momentum across the bottom boundary because scalar and velocity gradients go to 0. The von Neumann boundary conditions at the top of the domain correspond to the sub-grid fluxes of heat, salt and momentum (Equations 8, 14, and 15), as resolved fluxes go to zero at a no-penetration boundary. The roughness length ($z_0$ in Equations 8 and 12) is chosen such that the equivalent drag coefficient assuming quadratic drag is 0.003, an intermediate value for sea ice and ice shelf bottoms, though poorly constrained (Holland

and Jenkins, 1999; Holland and Feltham, 2006; MacAyeal, 1984; Nicholls et al., 2006). A list of parameter choices can be found in Table S1.

     We present two sets of simulations which have a base case in common. The base case has low far-field thermal driving of $0.15\,^\circ\mathrm{C}$ and a relatively high slope for Antarctic ice shelves of $1\,^\circ$ and vigorous far-field inertial oscillations of $20\,\mathrm{cm\,s}^{-1}$ generated by a $0.03\,\mathrm{Pa\,m}^{-1}$ pressure gradient. These conditions were chosen to favor an energetic regime, for reasons discussed

in Section 4.1. To examine the relationship between thermal driving and melt rate, as well as turbulent flow characteristics, we vary the far-field thermal driving from $0.15\,^\circ\mathrm{C}$ to $0.6\,^\circ\mathrm{C}$ in the first set of simulations. The far-field thermal driving, $\Delta\theta_\infty$, is defined as the difference between the far-field temperature ($\theta_\infty$) and the freezing temperature based on the far-field salinity ($\theta_f(S_\infty)$). In the second set of simulations, we reduce the slope from $1\,^\circ$ to $0.01\,^\circ$. The ice base always slopes to the east in the positive x-dimension of our domain, thus $\beta = 90\,^\circ$. The far-field salinity for all runs is $35\,\mathrm{g\,kg}^{-1}$. Initially, both temperature

and salinity increase with depth over the upper two-thirds of the domain. The background salinity stratification dominates the density stratification, with an inverse stability ratio $R_\rho^*$ of 20.





The simulation duration is 50 h, corresponding to ∼4 inertial periods of 13 h. By the end of our simulations, the time-mean kinetic energy of the flow has reached steady-state. However, the turbulent intensity for all cases continues to decline, with more pronounced turbulence kinetic energy (TKE) loss at lower slopes (Figure 1a,d). Unless stated otherwise, the results are
presented as averages over the last simulated inertial period and over the domain excluding the sponge layer.

We compute an effective thermal exchange coefficient that differs from that employed by the sub-grid scheme to represent the efficiency of heat exchange that may need to be represented to produce accurate melt rates in an ocean model that does not have the vertical resolution or sophisticated turbulence closure that we employ here. This derived thermal exchange coefficient, $\Gamma_{T,der}$, can be computed using Equations 10 and 16 from the simulated melt rate and ocean properties at any depth below $z_{\frac{1}{2}}$,
here chosen at -2 m. We substitute the friction velocity computed by Equation 8 with one computed using a quadratic drag law from the velocity simulated 2 m below the boundary and the applied drag coefficient, consistent with drag implementations in coarse-resolution ocean models.

## 3   Results

Melt rates decline over the course of the simulations (Figure 1), preventing the identification of steady-state melt rates for
most runs. This decline is in response to the concomitant increase in stratification during the course of the simulations, which decreases vertical heat fluxes by reducing vertical turbulent fluctuations. We do not continue our simulations beyond 4 inertial periods in the hope of reaching steady-state conditions because the increase in stratification reduces the turbulent length scales and necessitates higher model resolution than we could computationally afford. Our simulations generally show resolved turbulent fluxes exceeding subgrid turbulent fluxes by at least a factor of two, but subgrid turbulent fluxes dominate within
several meters of the ice base where the stratification is strongest (Figure S1). Resolved turbulent fluxes are also comparable to subgrid turbulent fluxes late in the simulation of low slope cases after significant TKE has been lost ($\leq 0.1°$C slopes, Figure S1d).

We present depth-profiles of temperature, salinity, and velocity at the end of all simulations (Figure 2). The time-averaged far-field velocity shown in Figures 2c,f removes a periodic signal from inertial oscillations of the geostrophic velocity with a
characteristic magnitude of $20\,\mathrm{cm\,s}^{-1}$. Ekman rotation near the boundary can be seen in all simulations (Figs. 2c,f), but for more strongly sloped runs, buoyancy plays an increasingly strong role in driving the mean flow near the boundary. This effect can be seen most clearly by comparing the up-slope component of the flow within several meters of the ice-ocean interface across the slope-varying simulations (Figure 2f). These mean buoyancy effects increase flow velocities on the order of a few $\mathrm{cm\,s}^{-1}$ as slope varies from $0.01°$ to $1°$. Flow velocities near the boundary increase on the order of $10\,\mathrm{cm\,s}^{-1}$ as far-field
thermal driving increases from 0.5 to $0.6°$C at $1°$ slope (Figure 2c). This velocity increase is attributed to changes in the magnitude of buoyancy forcing near the boundary, related to differences in melt rates. The vertical momentum flux profiles shown in Figure 3a,c reveal that flow is accelerated (negative fluxes) over much of the IOBL, with drag dominating only within the first few meters below the boundary.

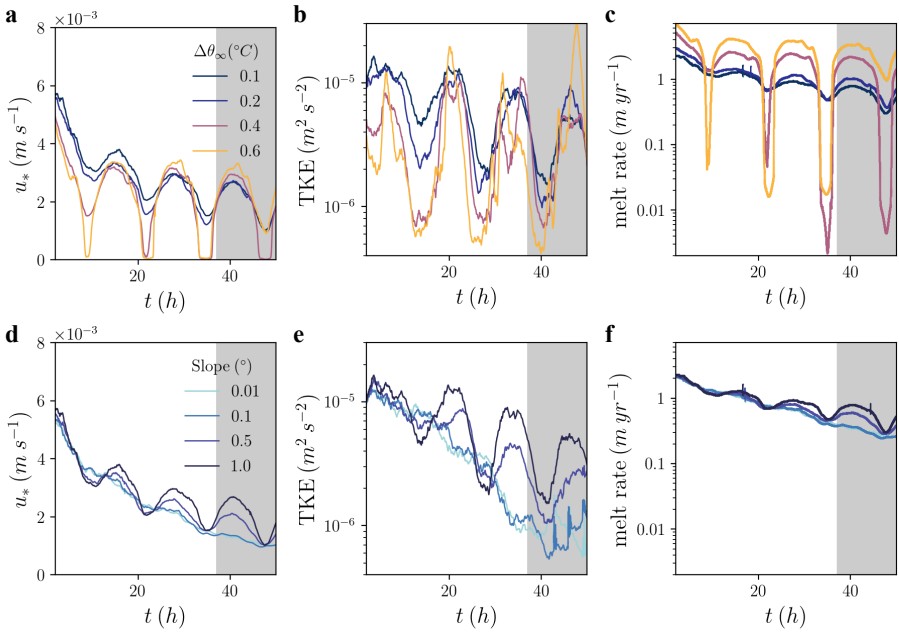

**Figure 1.** Time evolution of (a,d) domain-averaged, resolved turbulence kinetic energy (TKE), (b,e) melt rate and (c,f) friction velocity for (a-c) thermal driving simulations and (d-f) variable slope simulations. The black curve represents the same simulation in all panels in this and subsequent figures. The analysis window, the last inertial period, is shaded.

All simulations show an evolution from the weakly stratified initial conditions to more strongly stratified conditions, par-
ticularly within the first $5\,\mathrm{m}$ of the ice-ocean interface (Figures 2b,e). In none of the simulations do we observe a well-mixed boundary layer with respect to scalars. Rather, the simulations show varying degrees of stratification over the first $20\,\mathrm{m}$ from the boundary. Stratification within the boundary layer increases with thermal driving (Figure 2a,b). Thus, the effect on strati-
fication of the increase in melt-induced buoyancy fluxes with thermal driving dominates over the increase in shear induced by the buoyant flow. Conversely, stratification decreases with increasing slope, indicating that the increase in shear induced by the
buoyant flow dominates over the increase in melt-induced buoyancy fluxes with slope.

Shear production of turbulence dominates the TKE budget. The evolution of TKE can be described by three source terms and dissipation:

$$\frac{de}{dt} = \underbrace{-\overline{u'w'}\frac{du}{dz} - \overline{v'w'}\frac{dv}{dz}}_{F_{shear}} \underbrace{-\frac{g_3}{\rho}\overline{\rho'w'} - \frac{g_1}{\rho}\overline{\rho'u'}}_{F_{buoy}} + \underbrace{\frac{d}{dz}(\overline{w'p'} + \overline{w'e'})}_{F_{trans}} - \underbrace{\varepsilon}_{diss} \tag{19}$$

Here, we characterize the evolution of resolved TKE and primes designate resolved fluctuations. Figure 4b-d shows the
source terms in this budget for the variable slope simulations; the analogous figure for the thermal driving simulations is Figure S2 and shows similar patterns. Shear production ($F_{shear}$) ranges from $\sim 10^{-9}$–$10^{-8}\,\mathrm{m^2\,s^{-3}}$ with a maximum at the ice-

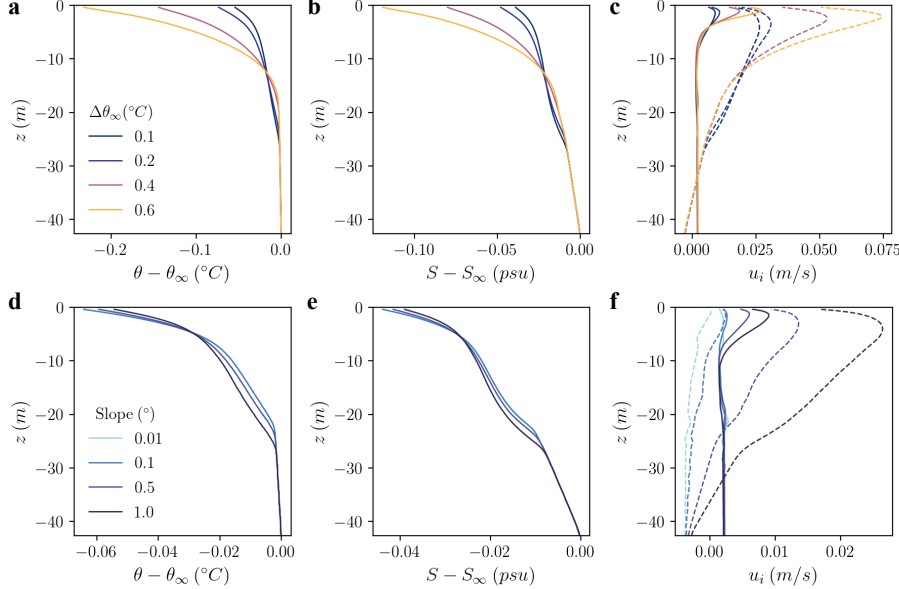

**Figure 2.** Depth profiles of simulated properties as they vary with (a-c) thermal driving and (d-f) slope, averaged over the last inertial period. (a,d) temperature relative to far-field temperature, (b,e) salinity relative to far-field salinity, and (c,f) velocity in the y-direction (solid line, positive north, cross-slope) and in x-direction (dashed line, positive east, up-slope).

ocean interface and local maxima in the upslope flow within $5\,\mathrm{m}$ of the boundary (Figure 4a). The increase in TKE throughout the boundary layer as slope increases reflects this shear-induced turbulence (Figure 4a).

Buoyancy production of turbulence ($F_{buoy}$) is $1-2$ orders of magnitude smaller than shear production, $\sim 10^{-10}$–$10^{-9}\,\mathrm{m}^2\,\mathrm{s}^{-3}$.
For a sloped ice shelf, the buoyancy term can be broken into two components, the horizontal buoyancy fluxes that increase turbulence when these fluxes are oriented upslope and the vertical buoyancy fluxes that decrease turbulence when ice is melting (Equation 19). Since our simulations only produce melting, the vertical component is always negative (dashed lines, Figure 4c). Over the coarse of an inertial oscillation, the horizontal component of buoyancy production is generally positive when the mean flow is oriented upslope and negative when the the flow is oriented downslope. Interestingly, the time-averaged effect of
a slope is destruction of turbulence near the boundary and production of turbulence near the base of the boundary layer (dotted lines, Figure 4c). The net effect of both horizontal and vertical buoyancy components is the destruction of turbulence throughout the IOBL (solid lines, Figure 4b), with the exception of the base of the IOBL where the horizontal buoyancy component augments entrainment. While these complexities are intriguing from a dynamical perspective, we do not explore them further here since the buoyancy term is not of leading-order in the TKE budget.

The transport term ($F_{trans}$) contains two components, advection of TKE due to pressure fluctuations and turbulent advection of TKE. The former is negligible, reaching maximum values on the order of $10^{-12}\,\mathrm{m}^2\,\mathrm{s}^{-3}$ while the latter increases turbulence at the boundary with oscillations of decreasing amplitude with distance from the boundary (Figure 4d).

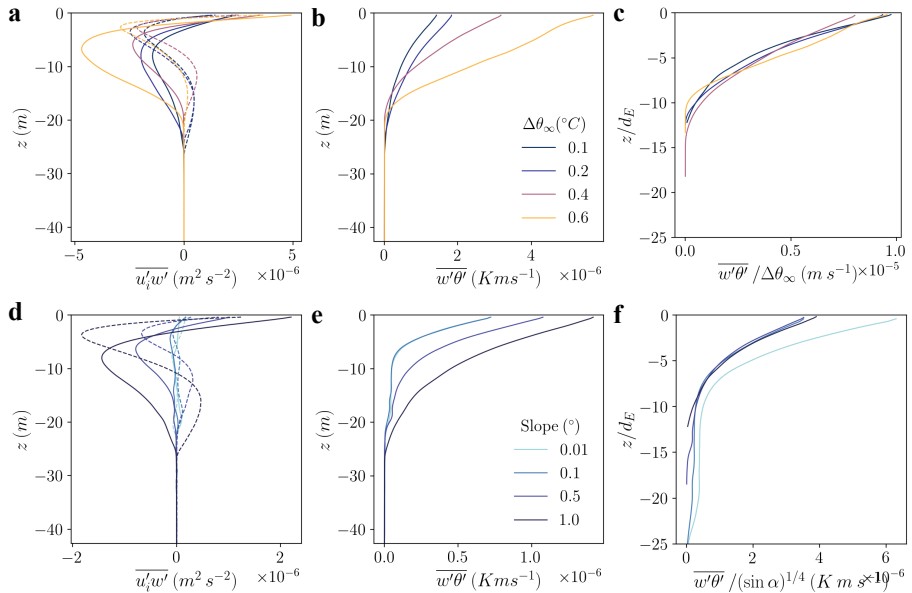

**Figure 3.** Vertical profiles of (a,c) momentum flux, (b,e) heat flux, and (c,f) scaled heat flux averaged over the last inertial period. The first row shows temperature cases, the second row shows slope cases. Momentum flux is expressed in two components: $\overline{u'w'}$ (dashed) and $\overline{v'w'}$ (solid). Positive flux denotes upward flux (i.e., drag). The horizontal axis limits vary between panels.

Dissipation ($diss$), the remaining term in the TKE budget, can be inferred from the remainder of these terms and the rate of change of TKE. Since we evaluate resolved TKE ($e$), dissipation here represents the transfer of energy to the sub-grid scales;
there is no flow of energy from sub-grid to resolved scales in the turbulence closure scheme. In these simulations TKE is not in steady-state, with an average dissipation rate over the course of a simulation of $\mathcal{O}(10^{-11})\,\mathrm{m^2\,s^{-3}}$ as seen in Figure 1a,c. Dissipation in the IOBL is on the same order as shear production ($10^{-9}\,\mathrm{m^2\,s^{-3}}$), as all other terms in the TKE budget are small.

To demonstrate the simulated turbulent structures in this regime, we present horizontal and vertical cross-sectional snapshots
through the domain for the two slope end-members in Figure 5. Turbulent structures within the IOBL are consistent with propagating Holmboe shear instabilities under stable stratification (Carpenter et al., 2010). Shear is stronger within the IOBL than at the base of the IOBL due to the concentration of buoyant plume flow near the top of the IOBL. Consequently, the amplitude of these structures increases near the boundary for the more strongly-sloped simulations (Figures 5). The difference in IOBL turbulence with slope is perhaps best seen in the turbulent structures at $1\,\mathrm{m}$ from the ice-ocean interface (Figure
5e,f). The structures become increasingly filamentous (i.e., near-wall streaks, e.g., del Álamo and Jiménez (2003); Hoyas and Jiménez (2006)) and coherent as slope and thus the velocity of the buoyancy-driven current increase. Since the stratification decreases with increasing slope, the ratio of vertical to horizontal velocity variance also increases; velocity fluctuations are less confined to the slope-parallel direction (Figure S3).

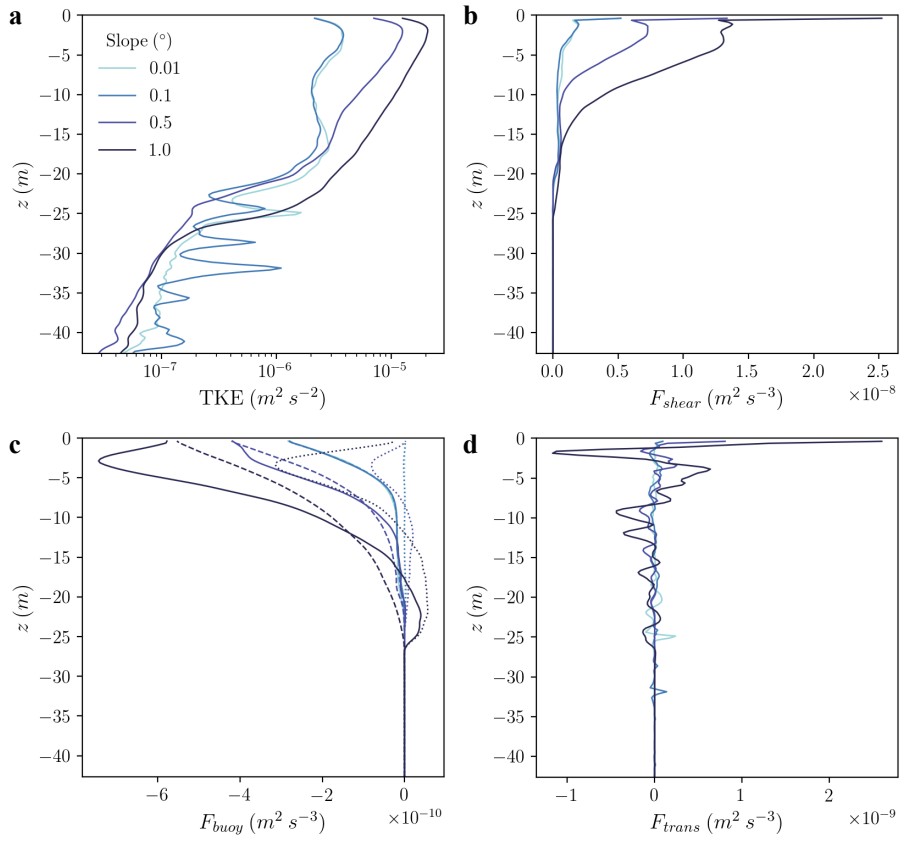

**Figure 4.** (a) Simulated turbulence kinetic energy (TKE) and (b-d) TKE source terms for variable slope simulations averaged over the last inertial period. (b) Shear production. (c) Buoyancy production: total (solid lines), vertical component (dashed), and upslope component (dotted). (d) TKE transport. Positive denotes production, negative destruction of TKE. Note that the x-axis scales differ between panels.

We define the base of the IOBL as the depth where the thermal driving relative to the far-field freezing point is 99% of

the far-field thermal driving. The IOBL depth increases through time, reaching $13 - 19\,\mathrm{m}$ at the end of the simulations. The simulated boundary layer depth increases with the far-field thermal driving (from $13$ to $19\,\mathrm{m}$) and with ice-shelf basal slope (from $15$ to $19\,\mathrm{m}$), reflecting the increase in flow velocities across those parameter changes which drives entrainment into the IOBL. Figure 6 shows the temporal evolution of IOBL depth for a few simulations, with steady IOBL growth at low slopes and punctuated growth at higher slopes corresponding to trends in TKE.

Given the temporal variability in TKE in these simulations, we use a threshold in dissipation rate to characterize the mixing layer depth, as distinct from the mixed layer depth. This criterion has been deployed for stratifying ocean boundary layers (Franks, 2015; Sutherland et al., 2014). We consider the mixing layer as the depth interval over which the horizontal- and hourly-averaged dissipation rate exceeds $10^{-9}\,\mathrm{m^2\,s^{-3}}$. This mixing layer depth shows greater temporal variability than the

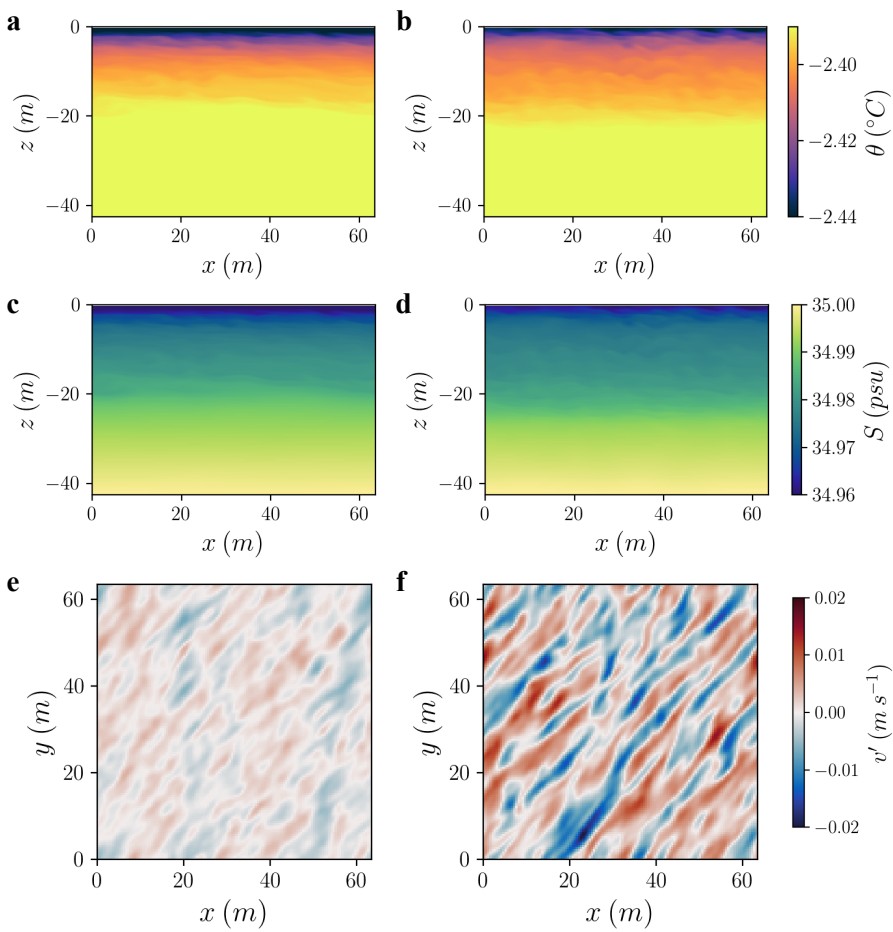

**Figure 5.** Instantaneous flow structures observed at 40h in (a,c,e) 0.01° slope case and (b,d,f) 1° slope case. (a,b) Temperature in cross section mid-way through the y-axis. (c,d) Salinity in cross-section mid-way through the y-axis. (e,f) Resolved cross-slope velocity fluctuations at 1 m below the ice-ocean interface.

IOBL depth as defined by scalar concentration, and drops below that IOBL depth during periods of enhanced entrainment

(Figure 6).

There are also time periods over which the dissipation rate drops below $10^{-9}\,\mathrm{m^2\,s^{-3}}$ throughout the water column, indicating intermittency in turbulence. We note that these intervals of low dissipation do not correspond to complete loss of TKE, nor a shoaling of the IOBL depth as we have defined it (Figure 6). At higher slopes ($\geq 0.5°$), this intermittency is interrupted as inertial oscillations enhance up-slope IOBL flow, increasing shear-driven mixing. This mixing front begins at the boundary

and propagates to the base of the IOBL over a few hours (Figure 6b). For the less stratified, low-slope cases, the intermittency in turbulence is more frequent, as the IOBL flows more slowly and generates less TKE production by shear (Figure 6a). We discuss this intermittency and its possible implications further in Section 4.1.





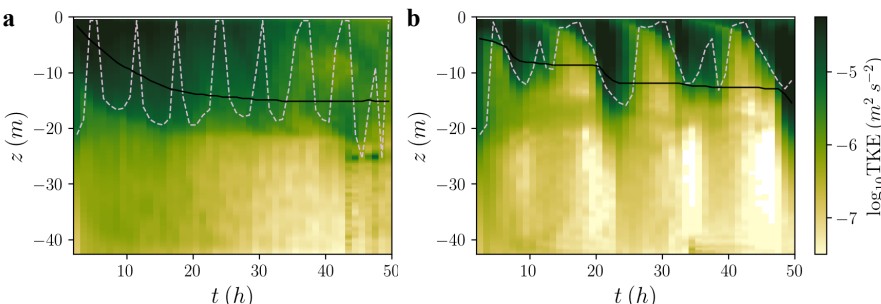

**Figure 6.** Horizontally-averaged turbulence characteristics for (a) $0.1\,^\circ$C thermal driving, $0.01^\circ$ slope case and (b) $0.6\,^\circ$C thermal driving, $1^\circ$ slope case. Turbulence is considered intermittent when the dissipation contour of $10^{-9}\,\mathrm{m^2\,s^{-3}}$ (dashed line) reaches the boundary. Significant turbulence kinetic energy (TKE, green-yellow shading) can be present when dissipation is low. Higher TKE at $-25\,\mathrm{m}$ at later times in (a) is due to turbulent transport, while the higher TKE at later times in (b) is due to shear production. IOBL depth is contoured in black.

The dominant temporal frequency in melt rate is the inertial frequency, and the melt response to those oscillations is highly nonlinear. Maximum melt rates occur when the mean flow is oriented between the up-slope direction and the Coriolis-favored direction, and minimum melt rates roughly $180^\circ$ opposed to that. These melt rate fluctuations correspond to fluctuations in TKE which are reflected in the friction velocity shown in Figure 1b. The dominant contribution to turbulence is shear production of TKE, which changes its distribution with depth as the mean flow profile evolves. During high-melt periods, the far-field flow is oriented up-slope and shear production of TKE is concentrated near the boundary. During low-melt periods, the far-field flow is oriented down-slope and shear production of TKE is concentrated a few meters away from the boundary. Melt rate fluctuations increase in amplitude as thermal driving increases and as slope increases, which we attribute to the increasing importance of buoyancy forcing in driving a near-boundary plume and thus determining the depth-distribution of shear. The two simulations associated with the highest thermal driving values, and the highest near-boundary stratification, experience a dramatic reduction in melt rates during the down-slope flow period. This coincides with reduced friction velocity (i.e., shear stress) at the ice interface (Figure 1b) and reduced vertical velocity fluctuations (not shown).

Time-averaged melt rates depend fairly linearly on far-field thermal driving (Figure 7a, $R^2 = 0.97$). This is mostly attributable to a linear relationship between melt rate and interfacial thermal driving, as differences in friction velocity and the thermal exchange coefficient are small and do not have a systematic relationship with far-field thermal driving. On the other hand, the derived, time-averaged thermal exchange coefficient representing the efficiency of heat exchange from $-2\,\mathrm{m}$ depth to the ice base ($\Gamma_{T,der}$) does have a weakly negative relationship with far-field thermal driving (Figure 7c). This indicates a decrease in the efficiency of heat exchange with increasing near-interface stratification. The highest thermal driving case shows an anomalously high thermal exchange coefficient over the last inertial period, which features high TKE shown in Figure 6b. We discuss these derived thermal exchange coefficients further in Section 4.2.2.

There is also a linear relationship between melt rate and ice-shelf basal slope, with threshold-like behavior at slopes less than $0.01^\circ$. This linear relationship is due primarily to a linear relationship between friction velocity and melt rate while

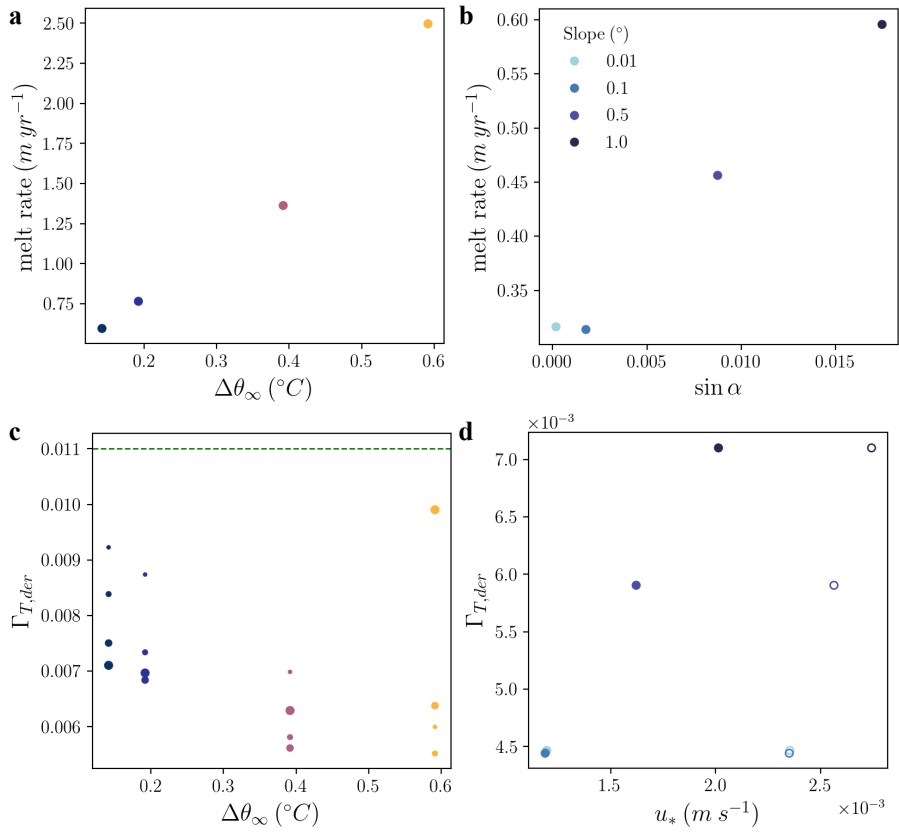

**Figure 7.** Melt rate sensitivity to (a) far-field thermal driving and (b) sine of the basal slope. (c) Far-field thermal driving is inversely related to $\Gamma_{T,der}$. Dashed line denotes the value recommended by Jenkins et al. (2010). The largest points correspond to the fourth inertial cycle with progressively smaller points for previous inertial cycles. (d) For variations in slope, the simulated friction velocity (solid points) is linearly related to $\Gamma_{T,der}$. The inferred friction velocity used to compute $\Gamma_{T,der}$ are shown with open points. Note the difference in y-axis limits from (c). The $0.01°$ and $0.1°$ slope cases are overlapping.

there is a smaller ($\sim 1/3$ size) opposite effect from decreasing interfacial thermal driving with increasing slope (not shown). These differences in friction velocity arise from higher IOBL velocities and turbulence at higher slopes. We observe threshold behavior in melting in the two lowest-slope cases, which can be attributed to similar friction velocities arising from similar IOBL velocities and turbulence (Figure 7d and Figure 2f). This behavior is discussed in more detail in Section 4.2.1. The differences in derived thermal exchange coefficient with varying slope are on the order of a 40% change as slope increases

from $0.01°$ to $1°$ (Figure 7d).

     Vertical heat fluxes are shown in Figure 3b,e. The vertical heat flux has a maximum at the ice-ocean interface and decreases throughout the IOBL, with small values below the pycnocline. Since the IOBL is fully turbulent (with the exception of some





intermittency discussed below), the sub-grid diffusivities of momentum, heat and salt closely resemble one another (Figure S4). The vertical salt flux profiles are shown in Figure S5 and have a very similar shape to the vertical heat flux profiles.

The relatively narrow range of conditions simulated here suggests that a depth-dependent shape function for scalar fluxes could be formulated. The distance from the interface is scaled by the Ekman depth:

$$d_E = (2K_e)^{1/2}|f_3|^{-1/2} \tag{20}$$

where $K_e$ is the mean eddy viscosity in the turbulent boundary layer assuming the total fluxes follow Fick's law:

$$\overline{w'\boldsymbol{u}'} = K_e d\boldsymbol{u}/dz \tag{21}$$

$K_e$ profiles are shown in Figure S6.

The linear scaling of melt rate with thermal driving suggests a linear scaling of heat flux profiles with thermal driving. We find that this scaling largely collapses the four thermal driving profiles, with notable deviation from this shape for the run that experiences temporal gaps in shear stress during the analysis period (pink curves in Figures 1c, 3c). The shape of these profiles can be reasonably approximated by a linear decrease in the scaled heat flux with scaled depth over the boundary layer (Figure 315    3c). Scalar fluxes decline near the boundary for the $1°$ slope cases, a feature that we discuss further in Section 4.2.3.

Despite melt rates scaling reasonably well with the basal slope $(\sin\alpha)$, the vertical heat flux profile does not, showing a much lower sensitivity to slope $((\sin\alpha)^{1/4}$, Figure 3f). There is strong agreement between the scaled vertical heat flux profiles at different slopes, though the threshold behavior at low slopes noted in melt rates is replicated here. We also find that the Ekman depth is a poor predictor of boundary layer depth. This may be due to the depth-variable shear induced by buoyancy 320    which is not reflected in the depth-mean IOBL eddy viscosity used to compute the Ekman depth. We discuss the scaling of vertical fluxes and the possibility of their parameterization in Section 4.2.3.

## 4   Discussion

### 4.1   Understanding IOBL turbulence and the limitations of an LES approach

In the simulations presented here, turbulence declines throughout the course of the simulation, becoming intermittent. The 325    relationship between stable stratification, shear, and the persistence of turbulence remains an open question (Zonta and Soldati, 2018). Thus, it is not possible a priori to determine whether the level of turbulence simulated by this LES model is appropriate for the regime space we have sampled. While we did conduct model validation against a stably stratified atmospheric boundary layer test case (see Section 2.2), the degree of stratification at the boundary in that case did not approach that simulated in the sub-ice configuration. Fundamentally, we cannot guarantee that our LES is not overly dissipative such that the TKE generated 330    by the resolved dynamics is lost too quickly relative to real-world sub-ice settings.

Excess dissipation could arise either through the sub-grid scheme or the model numerics. We found a rapid loss of turbulence in PALM simulations when a dynamic Smagorinky turbulence closure was used, which is consistent with previous studies on the limited applicability of the Smagorinsky turbulence closure to strongly stratified flows due to the strong anisotropy of





those flows (Flores and Riley, 2011; Jiménez and Cuxart, 2005). This motivated our adoption of the AMD turbulence closure

scheme (Abkar et al., 2016). However, we found that the buoyancy term added to this scheme by Abkar and Moin (2017) had

an unrealistically high magnitude in the vicinity of the ice base where gradients are large. We removed this term, as it was

negligible in the simulations of Vreugdenhil and Taylor (2019) (personal communication), but our unrealistic solution for this

term suggests that AMD scheme may not perform optimally at the resolution employed here. Though the resolution of our

simulations is significantly lower than that used by Vreugdenhil and Taylor (2019), it enables us to extend the domain from

their $2\,\mathrm{m}$ to $64\,\mathrm{m}$ to allow for the development of a thick IOBL.

Strongly stratified turbulence has been associated with intermittent turbulence (Nieuwstadt, 2005; Wiel et al., 2012), though

there are also numerical experiments that fail to produce intermittency even under strongly stable stratification (Arya, 1975;

Komori et al., 1983). This is not the first study to find the emergence of intermittent turbulence in stably stratified, sub-ice

settings (Vreugdenhil and Taylor, 2019). Donda et al. (2015) have argued that, in strongly stratified flows, the cessation of

turbulence is transient provided there are sufficiently large perturbations. This is consistent with our finding that the temporal

variability in shear over inertial oscillations provides sufficient perturbations to reinitiate turbulence. Our simulations approach

a gradient Richardson number ($\mathrm{Ri}_g$) of 0.25, which is considered to be the approximate value at which turbulence neither grows

nor decays (Holt et al., 1992; Rohr et al., 1988). Thus, fluctuations in TKE are plausible at the simulated levels of stratification

and shear. However, when turbulence is intermittent, as it is in this study, the application of LES may be inappropriate due to

its inherent horizontal averaging over laminar and turbulent regions (Stoll and Porté-Agel, 2008). Thus, our results should be

interpreted with caution.

In this study, we did not attempt to reproduce observed conditions at a particular ice-shelf location due to the difficulties

of matching unobserved far-field forcings and the exclusion of tides from our simulations. Nonetheless, it appears as though

well-mixed boundary layers are seen for a narrower range of conditions in simulations than in observations. The geostrophic

flow chosen in these simulations is quite strong at $20\,\mathrm{cm\,s^{-1}}$ and thermal driving is relatively low such that we expected to

produce a well-mixed boundary layer as is observed in melting regions of the Filchner-Ronne Ice Shelf (Nicholls et al., 2001).

However, observations to date are insufficient to fully characterize the regime space for a well-mixed IOBL (Malyarenko et al.,

2020). The observational picture is quite nuanced with a range of stratification observed even within one ice shelf (Hattermann

et al., 2012).

Shear-driven turbulence within the IOBL plays a central role in determining vertical heat fluxes and thus melt rates. In these

simulations the destruction of TKE by the stabilizing buoyancy flux is two orders of magnitude smaller than shear production

of TKE throughout the IOBL. Our finding that shear production of TKE dominates over the buoyancy term is consistent with

Davis and Nicholls (2019) who found that shear production was an order of magnitude greater than buoyancy destruction of

TKE in the IOBL below Larsen C Ice Shelf.

The flux Richardson number, $\mathrm{Ri}_f$, the ratio of buoyancy flux to shear-driven TKE production, provides a measure of mixing

efficiency. The simulated $\mathrm{Ri}_f$ values of $0.05 - 0.1$ are well below the critical $\mathrm{Ri}_f$ of $\sim 0.25$, indicating that we are in the

regime in which mixing efficiency ($\mathrm{Ri}_f$) decreases with increasing stratification ($\mathrm{Ri}_g$) (Armenio and Sarkar, 2002; Peltier and



Caulfield, 2003). This is consistent with our finding that the derived, $2\,\mathrm{m}$-depth thermal exchange coefficient decreases for the high thermal driving cases which also achieve stronger IOBL stratification.

Our simulations are certainly missing some sources of TKE present in ice-shelf cavities which could modify IOBL structure and mixing efficiency. These simulations did not include tides, which provide perturbations to the mean velocity that enhance melt rates and entrainment, especially at Filchner-Ronne (Makinson and Nicholls, 1999; Makinson et al., 2011; Mueller et al., 2018). While internal gravity waves arise in LES, in our LES model they may be of smaller amplitude and play a lesser role in mixing than they would in a real ice-shelf settings due to the absence of large-scale external forcings such as tides and

storms, seafloor topography, as well as possible resonance with the cavity geometry (Gwyther et al., 2020; Mueller et al., 2012; Padman et al., 2018; Robertson, 2013). Enhanced drag at the ice-shelf base could increase shear production of TKE; however under strong stratification, surface roughness elements may suppress turbulence rather than enhance it (Ohya, 2001).

## 4.2   Representing the IOBL and projecting ice-shelf melt rates with ocean models

### 4.2.1   Insights into melt rate sensitivity to ocean conditions

The decline in IOBL turbulence discussed in Section 4.1 results in declining melt rates. Thus, we could not evaluate the relationship between melting and far-field conditions at steady state, which would have offered the most direct path to assessing melt rate sensitivity to ocean conditions. As discussed in Jenkins (2016), achieving steady-state solutions in simulations may require prescribing large-scale gradients in temperature. We have not included these large-scale gradients in our simulations, but this may be an avenue for future work. However, we believe that our transient solutions do provide some indications of

how melt rates and boundary layer properties depend on ocean temperature and ice-shelf slopes. One justification for this belief is that the simulated sensitivity of melt rate to ocean temperature remains consistently linear through all inertial periods simulated (Figure S7). On the other hand, the differently sloped simulations continue to diverge at the end of the simulation as the IOBL accelerates (Figure 1e,f). Nonetheless we are able to find relationships between the vertical heat flux profiles and ocean temperature and slope that suggest predictability of the mean effects of turbulence despite the transience of these

simulations.

   The linear relationship we find between local, far-field ocean temperature and melt rates is consistent with some previous studies in ice-shelf settings (Holland et al., 2008; Rignot and Jacobs, 2002; Vreugdenhil and Taylor, 2019). A slightly higher exponent of $4/3$ is also consistent with our data ($R^2 = 0.95$); a value reported for the regime in which convective instabilities control melting while our simulations feature shear instabilities (Kerr and McConnochie, 2015). In contrast, in sea-ice settings

this sensitivity of melt rates to local thermal driving was found to be significantly smaller with an exponent of 0.38 (Ramudu et al., 2018). It's worth noting that our simulations do not capture the large scale increase in overturning circulation that accompanies a distant increase in thermal driving (i.e., changes to the water masses entering the ice-shelf cavity). The relationship between melt rate and distant thermal driving is a function of both the relationship between the local thermal driving and melt rate and the relationship between distant thermal driving and friction velocity (Holland et al., 2008). The former relationship

is addressed by this study; the latter is only partially captured by the increase in IOBL velocity due to an increase in buoyancy





while the far-field velocity remains constant. We note that the often cited quadratic relationship between thermal driving and melting pertains to this distant thermal driving (Holland et al., 2008); studies generally find this exponent to be between 1.5 and 2 (Favier et al., 2019; Jourdain et al., 2017; Little et al., 2009).

To some extent, this linear relationship between far-field thermal driving and melt rate is embedded in the parameterization
of heat fluxes employed at the ice base in these simulations. Specifically, the melt rate is prescribed to have a linear dependence on the *local* thermal driving, based on the temperature in the first model layer (Equation 10). However, the level of stratification near the ice base, the buoyant acceleration of the IOBL plume and the transport of heat from depth to the IOBL all mediate the relationship between *far-field* thermal driving and melt rates, and yet the dependence of melt rate on far-field thermal driving from these simulations is still fairly linear.

The relationship between melt rate and ice-shelf basal slope ($m \propto (\sin \alpha)^n$) combines two effects: the effect of ice-shelf slope on the IOBL's mean velocity profile and the effect of ice-shelf slope on IOBL turbulence. In the three-equation parameterization of ice-shelf melting, the former has the strongest effect on the friction velocity as derived from the mean flow velocity at a given depth; whereas the latter is represented by the scalar exchange coefficients, with higher values indicating more efficient turbulent transport. We return to the implications of our study for exchange coefficients in Section 4.2.2. As noted previously
in this section, while our simulations capture some of this IOBL acceleration, they do not reach a steady-state mean velocity profile. Thus, we cannot fully assess the the effect of ice-shelf slope on the IOBL's mean velocity profile. This effect is addressed by Magorrian and Wells (2016), whose scaling analysis predicted $n = 3/2$, and Little et al. (2009), who found $n = 0.94$ for the range of slopes considered here. Our finding that $n = 1$ may be applicable to the shear-dominated regime, in contrast to the sensitivity of $n = 2/3$ found in the convection-dominated regime at higher slopes than those simulated here ($5° - 90°$;
McConnochie and Kerr, 2018; Mondal et al., 2019). However, we emphasize that further investigation is needed beyond the small number of simulations presented here to validate our results.

The threshold-like behavior in melt rates at very low slopes is not predicted by geostrophic balance between Coriolis and buoyancy forcing, which dictates a linear relationship between $\sin \alpha$ and IOBL velocity (Jenkins, 2016), nor the scaling analysis of Magorrian and Wells (2016). This threshold behavior could be produced if any additional buoyant acceleration produced
by the small increase in slope from $0.01°$ to $0.1°$ increases both shear and dissipation, resulting in a negative feedback on IOBL turbulence. This is not evident in our simulations, which do not show significant differences in TKE budgets between the two runs (Figure 4). From a dynamical perspective, this may be an interesting target for more highly resolved LES in the near-boundary region. However, this threshold is located at low enough slopes that it is likely not of significance to melt rate parameterization for coarse-resolution ocean models, so we do not devote further attention to it here.

**4.2.2   Toward non-constant exchange coefficients in melt parameterization**

The thermal exchange coefficient as computed using Equation 11 at $0.125 \, \mathrm{m}$ below the ice-ocean interface (the uppermost grid cell) differs from that which would be implemented by coarse-resolution models or derived from oceanographic observations, both of which only know ocean properties meters to tens of $\mathrm{m}$ below the ice-ocean interface. To demonstrate the implications of this study for modeling endeavors, we computed the thermal exchange coefficient that yields the simulated melt rate using ocean




properties (temperature and velocity) 2 m below the ice-ocean interface. This depth was chosen to capture the faster portions of
       IOBL flow, representing a best-case scenario for high resolution ocean models, though this depth choice is somewhat arbitrary.

       These derived thermal exchange coefficients are shown in Figure 7c,d. The derived thermal exchange coefficients are all
       less than the value of 0.011 derived from observations (Jenkins et al., 2010). There are two main factors that contribute to
       this result. The first is the choice of the parameterization of fluxes at the ice-ocean interface (i.e., over sub-meter scales). We

chose a stability-dependent parameterization in which the buoyancy forcing from melting enters through the Monin-Obukhov
       length (Equation 9). We believe there is a stronger case for the flux parameterization we implement than for a constant thermal
       exchange coefficient in light of the success of Monin-Obukhov Similarity Theory (Monin and Obukhov, 1954; McPhee, 2008),
       and the depth-dependence of scalar fluxes in a more highly resolved sub-ice LES (Vreugdenhil and Taylor, 2019). Conse-
       quently, Vreugdenhil and Taylor (2019) also found that thermal exchange coefficients were less than the Jenkins et al. (2010)

value for all but the lowest thermal driving case. However, we acknowledge that more validation of the sub-grid boundary
       flux parameterizations is needed. The second contributing factor is declining simulated TKE, which reduces thermal exchange
       coefficients over the course of these simulations (Figure 7c).

       Our results also suggest a modest decline in the thermal exchange coefficient at higher thermal driving. This may lead to
       a sub-linear relationship between thermal driving and melt rate at higher thermal driving values, though this not evident in

our simulations. The decline in exchange coefficient with thermal driving agrees with Vreugdenhil and Taylor (2019), though
       their sensitivity is greater than that seen in this study (their Figure 8). As shown in Figure 7c, this relationship holds during all
       inertial periods, with an exception during an interval of strong turbulence (see Section 3). Due to the limitations of our LES
       modeling discussed in Section 4.1, we cannot recommend a best fit relationship between thermal exchange coefficient and
       thermal driving. Given the climatic importance of accurately simulating high thermal driving regimes associated with dynamic

ice-shelf thinning, LES coupled with observational validation across thermal driving regimes may be a fruitful avenue for
       future work.

       We find a linear relationship between the derived thermal exchange coefficient and the slope of the ice-shelf base, indicating
       that mixing is more efficient at higher slopes. This relationship also holds for the derived friction velocity from a quadratic
       drag law (Figure 7d). Note that these quadratic-drag friction velocities are greater than the model's parameterization of friction

velocity as the former neglects stratification effects while the latter includes them. This enhanced turbulent efficiency is due
       to enhanced shear instabilities, and is reflected in turbulent diffusivities parameterized by the AMD scheme. Much greater
       variability in sub-grid turbulent diffusivities is seen over the variation in slope than the variation in thermal driving tested here
       (Figure S4).

       The changes in the thermal exchange coefficient with slope were relatively small, from 0.0045 to 0.007 between 0.1° and

1° cases. For a coarse-resolution simulation, we anticipate that the failure to capture this slope sensitivity will not be a leading
       source of error in melt projections. Reproducing an accurate friction velocity is likely of a greater concern. Thus, the burden
       of melt projection accuracy may fall more heavily on parameterizing or resolving buoyant flow than on improving the slope
       dependence in the parameterization of scalar fluxes (i.e., improving Equation 10).





These simulations do not reveal whether a similar, linear relationship between friction velocity and thermal exchange co-
efficient holds when the background flow is varied rather than the slope. LES of sea ice melting, for which there is no slope,
suggests a sublinear relationship between thermal exchange coefficient and friction velocity ($\Gamma_T \propto u_*^{0.5}$, $\overline{w'\theta'} \propto u_*^{1.5}$; Ramudu
et al., 2018). More studies are required to determine this scaling.

### 4.2.3    Toward a vertical mixing scheme for the IOBL

There is reason to believe that improving ice-shelf basal melt projections using ocean models will require not only an accurate
melt parameterization but also an improved vertical mixing scheme. Jenkins (2021) demonstrated significantly different IOBL
characteristics when KPP is the turbulence closure scheme in contrast to a low- or high-order scheme. The eddy viscosity
simulated by our LES model shows better agreement with the viscosity solutions from Jenkins (2021) employing the low- and
high-order turbulence closure schemes than that employing KPP (compare our Figure S6 with his Figure 3). Thus, this study
offers additional support for the use of a more sophisticated turbulence closure scheme or a modified KPP scheme in sub-ice
settings, such as one based on the depth-dependence of vertical turbulent fluxes presented here (Figure 3).

A complication in melt parameterization of slope effects is the differential ability of ocean models and their vertical grid
configurations to capture IOBL flow. We find that this buoyant flow is concentrated in the uppermost 10 m, with peak velocities
as close as 3 m from the interface. This flow is unlikely to be resolved by most ocean models, which have typical vertical
resolutions near the interface of 10s m though some model configurations are reaching ∼1 m resolutions (Gwyther et al., 2020).
If the buoyancy-driven boundary current is unresolved in most coarse-resolution ocean models, then the friction velocity as
computed from a quadratic drag law and the velocity at the first grid cell is likely to underestimate the true friction velocity. For
instance, a simulation that lacks this buoyancy-induced current may look more like our negligible (0.01°) slope simulations
than those with a slope, and consequently underestimate melt rates. Thus, more accurate melt rate projections may result from
employing a depth-dependent parameterization of vertical scalar and momentum fluxes to provide some sensitivity of melt
rates to model resolution.

Our simulated depth-dependence for vertical scalar fluxes agrees with the shape of the turbulent scalar flux parameterization
of McPhee et al. (1987), which features a linear decrease in fluxes with distance from the boundary within the IOBL. The
low thermal-driving simulations with slopes of 1° diverge from this linear behavior within a few m from the boundary. We
attribute this to boundary effects that reduce TKE and eddy viscosity close to the boundary, reminiscent of the constant flux
layer predicted by Monin-Obukhov Similarity Theory (Monin and Obukhov, 1954). It is unclear whether a constant flux layer
is absent under some conditions or whether our simulations do not resolve it for those highly-stratified cases. Since these
differences in eddy viscosity are incorporated into the Ekman depth scaling, the scaled heat flux profiles all show quasi-linear
slopes (Figure 3c,f). This linear depth-scaling for IOBL fluxes should be explored under a broader range of conditions and at
higher resolution to determine whether it is broadly applicable and whether vertical mixing parameterizations need to improve
their representation of the constant flux layer in the IOBL.

A few complexities are immediately apparent in pursuing a depth-dependent shape function for vertical scalar flux param-
eterization. One is that a functional form for the eddy diffusivity is also needed to compute the Ekman depth. Another is that





any implementation of these depth-dependent scalar fluxes necessitates an inequality between the vertical heat and salt fluxes
from the top model grid cell and the heat and salt fluxes associated with melting. Thus, the closure of those budgets will likely
involve horizontal scalar fluxes as well.

A depth-dependent function for momentum is also necessary in ice-shelf settings given that strong stratification causes mo-
mentum fluxes to significantly deviate from the quadratic drag function typically employed in ocean models. Near-interface
stratification significantly decreases drag (García-Villalba and del Álamo, 2011; McPhee et al., 2008). Tuning the drag coeffi-
cient to fit present-day melt rates neglects the functional dependence of drag on melt rate through stratification (as represented
by the Monin-Obukhov length). The presence of an ice-shelf slope in ocean models that do not fully resolve a buoyant plume
further complicates the parameterization of momentum fluxes at the ice-ocean interface. A depth-dependent shape function
for momentum fluxes which may prescribe negative (downward) momentum fluxes at the top grid cell below ice shelves for
typical ocean model resolutions (see Figure 3a,d at, e.g., 5 m depth). In other words, when the buoyant plume is unresolved, the
momentum boundary condition may need to accelerate the flow at the boundary (at least in the up-slope direction) to produce
more accurate shear-driven mixing in the resolved portion of the domain.

## 5   Conclusions

In this study we presented a small parameter study of IOBL turbulence as captured by an LES model and chiefly explored
the sensitivity of ice-shelf melting to ocean temperature and ice-shelf basal slope. To our knowledge, this is the first study
to explore turbulent effects of low slopes on ice melting at scales spanning 10s m from the ice base, allowing for IOBL
development. These vertical scales allow us to examine the depth-dependence of vertical turbulent fluxes, a key component
for the development and validation of vertical mixing parameterizations in ice-shelf settings. Building on Jenkins (2021), our
results suggest an improvement on the standard KPP vertical mixing scheme is needed, but a larger simulation campaign and
field validation effort would be required to place such a scheme on strong footing.

The thermal exchange coefficient is a key parameter that determines the sensitivity of ice-shelf melting to changing ocean
conditions in ocean models. While a constant value is not theoretically supported, it remains the standard choice due to a lack of
consensus regarding the appropriate scaling. We concur with a previous study that the thermal exchange coefficient is dependent
on thermal driving such that melt rates are less sensitive to changes in ocean temperature (Vreugdenhil and Taylor, 2019). We
also suggest that the thermal exchange coefficient should be enhanced at higher slopes to account for shear instabilities that are
unresolved by coarse-resolution ocean models, with the caveat that our simulations did not reach steady-state.
Ultimately, we acknowledge that there is currently no universal guidance for parameterizing sub-grid scalar and momentum
fluxes near the ice base for ocean models, each of which capture IOBL flow differently depending on resolution, vertical
coordinate, numerical representations of the pressure gradient, and likely other aspects of the implementation (Gwyther et al.,
2020). Thus, progress toward accurate ice-shelf melt projections will require not only theoretical advances in understanding
melt rate sensitivity but also advances in numerical modeling to bring resolution- (scale-) awareness to melt parameterization.



*Code availability.* The modified version of PALM used for this study is currently under review at Los Alamos National Laboratory for open access release.

*Author contributions.* All authors conceived and designed the study. Begeman conducted and analyzed the simulations with input from Asay-Davis and Van Roekel. Begeman wrote the manuscript with input from Asay-Davis and Van Roekel.

*Competing interests.* The authors declare that they have no conflict of interest.

*Acknowledgements.* This research was supported by the Los Alamos National Laboratory (LANL) through its Center for Space and Earth Science (CSES). CSES is funded by LANL's Laboratory Directed Research and Development (LDRD) program under project number 20210528CR. This research was also supported by the LDRD program of LANL under project numbers 20210289ER and 20180549ECR. Support for simulation data analysis pertaining to ice-shelf melt parameterization and manuscript preparation was provided through the Scientific Discovery through Advanced Computing (SciDAC) program funded by the US Department of Energy (DOE), Office of Science,
Advanced Scientific Computing Research and Biological and Environmental Research Programs. Computing resources were provided by the LANL Institutional Computing Program, which is supported by the U.S. DOE National Nuclear Security Administration under Contract No. 89233218CNA000001.




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
