# Peer review of "Ice-shelf ocean boundary layer dynamics from large-eddy simulations"

_The Cryosphere, 2021_

## Author Response (AR1)

**Response to reviewers for "Ice-shelf ocean boundary layer dynamics from large-eddy simulations"**

Carolyn Branecky Begeman[1], Xylar Asay-Davis[1], and Luke Van Roekel[1]

[1]Los Alamos National Laboratory, P.O. Box 1663, Los Alamos, New Mexico, USA 87545

**Correspondence:** Carolyn Begeman (cbegeman@lanl.gov)

We thank both reviewers for their thoughtful consideration of our manuscript. We hope that our revisions, detailed below, have addressed your concerns and improved the manuscript.

**1  Response to Reviewer 1**

Abstract:

Line 2: "Yet these small scale processes, which regulate heat transfer between. . . ." Should be heat and salt transfer.
*The suggested change has been made.*

Introduction:

Line 27: ". . . .an overturning circulation known as "ice pump" Could talk about the refreezing process.
*We have added the following sentence to this paragraph:*

"Supercooling of the IOBL and frazil ice accretion to the ice-shelf base are also regionally important processes for cold cavities, but are not the focus of this work (Galton-Fenzi et al., 2012; Jordan et al., 2015)."

Line 31: " IOBLs present unique conditions in the global ocean, involving a stabilizing flux from phase change" The stabilizing flux mostly comes from the freshening of the boundary layer not from 'phase-change'.
*We intended to specifically refer to the stabilizing buoyancy flux at the boundary associated with ice-shelf melting. We hope this revision is more clear:*

"IOBLs present unique conditions in the global ocean, involving a stabilizing buoyancy flux from melting ice and a boundary layer that is positively buoyant against a sloping boundary."

Line 36: " Numerical studies addressing intermediate scale" Please define 'Intermediate scale'. Also to my knowledge Mondal et al., 2019 was a DNS study that resolved the Kolmogorov scale.
*We have avoided this vague term in our revision. The revised sentence reads:*

"IOBL turbulence has been explored through laboratory experiments, direct numerical simulations, and large-eddy simulations (Middleton et al., 2021; Mondal et al., 2019; Vreugdenhil and Taylor, 2019; McConnochie and Kerr, 2018)."

should clearly mention the author used a Lewis number =1.

*This comment wasn't entirely clear to us. If the reviewer is referring to this study, it isn't the case that the Lewis number=1 either in the melt parameterization or the AMD turbulence closure model.*

It would be interesting to see how friction length changes over slope, assuming eddy viscousity is estimated as a product of friction velocity and mixing length.

*Unfortunately, we have not output the 2-d friction velocity and momentum flux fields at the ice boundary necessary to compute a friction length as you define it. Furthermore, since the momentum fluxes at the boundary is defined by our subgrid near-wall scheme (Equation 8), you could express the friction length as $\frac{\kappa}{\Delta z}(ln\left(z_{\frac{1}{2}}/z_0\right) - \Psi_m\left(\zeta_{\frac{1}{2}}\right))^{-1}$. Thus, the friction length is chiefly a function of the Monin-Obukhov length. We can show how the Monin-Obukhov length varies with slope, as well as the ratio of resolved TKE 2m from the boundary to the friction velocity used by the near-wall scheme (figure below). However, we interpret these relationships to be primarily driven by the relationships between slope, mean velocity (and TKE) of IOBL, and melt rate already discussed in the manuscript.*

[Figure]

**Figure 1.** Relationship between Monin-Obukov length and the ratio of TKE at 2m below the boundary to the friction velocity evaluated at the boundary. All quantities are horizontally-averaged and time-averaged over the last inertial period.

Methods:

General Comment: I would love a schematic of the model set up.

*We have added a schematic figure, now labeled Figure 1.*

Overview of the LES model: It would help the readers if you clearly mention whether you have used a linear or non-linear equation of state.

*Thanks for catching the omission. We have added the following text:*

"We use the nonlinear equation of state from Jackett et al. (2006) to compute densities."

Simulation set-up:

Line 150: What about the time step?

*We have added the following text:*

"PALM employs adaptive timestepping; timesteps for the simulations presented here range from 0.5 s to 2.75 s after 2 h."

*We also realized that the following information might be useful to the reader:*

"We initiate turbulence over first 50 min of the simulation with perturbations to the horizontal velocity components on the order of 0.01 m/s."

Line 167: I don't think 1 degree is a relatively high slope for Antarctic ice-shelves.

*That's fair. We have deleted the judgment so that it now reads:*

"and a slope of $1°$"

Result:

A plot with temporal evolution of melt rate, obukhov lengthscales could help the reader.

*We have included a plot with the temporal evolution of melt rates, Figure 1 c,f. We believe that the obukhov lengthscale doesn't provide much additional context beyond that given by the melt rate and friction velocity in Figure 1, as it is computed from them. However, we added the following text to provide some context for the reader:*

"Average melt rates over the last inertial period range from 0.3–2.5 m (Fig. 2a,d) and average Monin-Obukhov lengthscales are $3$–$5\,\mathrm{m}$ (not shown)."

Discussion:

Line 397: 'The relationship between melt rate and distant thermal driving' : the sentence is hard to read.

*To make our point clear, we have rewritten the latter portion of this paragraph:*

"The relationship between average ice-shelf melt rates and the thermal driving for the cavity as a whole can be conceptualized in two components. The first is the relationship between the local thermal driving and melt rate, determined primarily by the local ocean turbulence. The second is the relationship between distant thermal driving (i.e., the water masses entering the ice-shelf cavity) and the strength of sub-ice-shelf circulation (Holland et al., 2008). Only the former is addressed by this study. Our simulations do not capture the large-scale increase in overturning circulation that accompanies an increase in distant thermal driving. Our simulations partially capture the increase in IOBL velocity due to an increase in thermal driving but the far-field velocity is unaffected. We note that the often cited quadratic relationship between distant thermal driving and melt rate involves both local turbulent processes and large-scale processes (Holland et al., 2008); studies generally find this exponent on distant thermal driving to be between 1.5 and 2 (Favier et al., 2019; Jourdain et al., 2017; Little et al., 2009)."

**2   Response to Reviewer 2**

This manuscript addresses boundary layer dynamics at the ice-ocean interface motivated by obtaining novel parameterizations of ice-shelf melting. This is a highly important topic: increasing complexity of ice and ocean models will not improve the estimates of melt rates with inappropriate representation of the processes in the boundary layer. Although the method chosen in the paper (LES using relatively low resolution) gives the results of limited applicability, which is admitted by the authors, it provides some important insights. The manuscript can be published if the authors address the following points

Major points:

Section 2.3: Please add the figure illustrating simulation setup for your base case. Add more detail in this figure: currently even the sizes of the domain are not written anywhere.
*We have added a schematic figure (Figure 1). This text specifying the domain size was included in the original manuscript, but it would be easy to miss. With the new figure, it should be more visible.*

"The domain is a 64 m$^3$ cube"

[Figure]

**Figure 1.** Schematic of the simulated ocean domain with background pressure gradient dp/dy. Purple arrow is oriented north and green arrow is aligned with gravitational acceleration. *The bottom boundary condition is Dirichlet, but there is also no flux as a result of damping.

L 100: please add more detail about validation in the appendix. It would be useful to have an idea how the chosen model works compared to other model (which one? Was it DNS or measurements or another LES model?).

*We have added a figure to the supplemental materials (Figure S1) and the following text to the manuscript. We don't feel that too much information is needed about model set-up because we follow Abkar and Moin (2017) exactly.*

90   "We validated our implementation against the stable atmospheric boundary layer test case published in Abkar and Moin (2017) and Stoll and Porté-Agel (2008). We follow Abkar and Moin (2017) model set-up exactly and chose their moderate resolution with 72x72x72 grid points. Our results are in good agreement with the Abkar and Moin (2017) solution employing another LES model with AMD (boundary layer height differs by <10%) and other LES models with different turbulence closures (Fig. S1, their Fig. 1 and references therein)."

[Figure]

**Figure S1.** Results of the atmospheric stable boundary layer test case presented in Abkar and Moin (2017) using PALM with the Anisotropic Minimum Dissipation (AMD) turbulence closure. Compare with their Figure 1.

95   L 150: How does the resolution influence the results? Why this particular resolution was chosen?

*We chose this resolution as the highest we could afford for this parameter study while achieving similar behavior to higher resolution simulations. We conducted one simulation at twice the resolution and compared the results, now described in the following text and a supplemental figure. We only ran the higher resolution simulation for one inertial period after spin-up due to the availability of computational resources.*

100   "We evaluated the effects of doubling both horizontal and vertical resolution in a separate simulation with an otherwise identical set-up to the base case, run for one inertial period after spin-up. While a greater portion of the vertical heat fluxes was resolved as expected, the melt rate averaged over one inertial period only increased by 7% and differences in the mean state were sufficiently small to justify the use of our standard resolution (Figure S2)."

[Figure]

**Figure S3.** Sensitivity of simulated mean state and turbulent fluxes to resolution. The vertical resolution is double the horizontal resolution $(\Delta x, \Delta y)$. Results are averaged over the first inertial period after a 2h spin-up. (a) Temperature. (b) Salinity. (c) Total vertical heat flux (solid), resolved vertical heat flux (dashed), and sub-grid vertical heat flux (dotted). (d) Velocity, u-component (solid) and v-component (dashed).

L 157-158: It would be helpful to write the boundary conditions explicitly, not just Dirichlet/von Neumann.

*We have modified this paragraph to explicitly state the boundary conditions, and rearrange some of the text accordingly. The relevant sentences are:*

"We set von Neumann boundary conditions at the top boundary corresponding to the dynamic sub-grid momentum and scalar fluxes (Equations 8, 14, and 15, as resolved fluxes go to zero at a no-penetration boundary. ... Boundary conditions are Dirichlet at the bottom boundary, set to the far-field temperature, salinity, and geostrophic velocity. ... The flow is periodic along the x and y dimensions."

If there is a flow developing along the ice face due to buoyancy, how justified is the use of the periodic boundary conditions?

*In the literature, periodic boundary conditions in the context of simulations of the IOBL have been employed either to represent the domain moving with the IOBL in a Lagrangian sense (Chen et al. 2021. The Cryosphere Discussions.) or the domain remaining fixed in space with periodic boundary conditions that provide the opportunity for the development of turbulence and of equilibrium conditions (Vreugdenhil and Taylor, 2019; Rosevear et al., 2021). We take the latter approach, with the intent to bring the flow to equilibrium with the forcing by the end of the simulations. As discussed in Section 4.1, equilibrium was not achieved.*

The whole Sec. 2.2, called 'Turbulence closure', is about boundary conditions at the ice-ocean interface.

*We agree that this may not be the clearest descriptor of this section. We move the first paragraph to the previous section and rename this section "Boundary conditions for ice-shelf melting."*

Section Results should be split into smaller subsections with separate titles to make it more reader-friendly.

*Good suggestion. We have added the following subsections: Overview of the mean simulated state, Turbulent kinetic energy budget, Boundary layer turbulence, Melting and its relation to thermal driving and slope, Vertical structure of turbulent fluxes.*

Minor points:

Please check the equations. Several equations have misprints:

*Thank you for reviewing the equations so carefully. We have made the changes you requested*

Eq (1), rhs: 1st term: should be $x_j$ and not $x_i$; 3rd term: what does index g in $u_g$ mean?

*We have added the following text to define $u_g$:*

"The momentum terms on the right hand side of Equation 1 are, in order: advection, Coriolis forcing, imposed geostrophic flow with a geostrophic velocity $u_g$, ..."

Eq (3),(4): 2nd terms in the rhs: should be $u_j$ not $u_i$;

*Change has been made.*

Eq (11): not clear, does this mean that gamma is the sum of the molecular and turbulent coefficients?

*The text has been modified to make this equation more clear:*

135 "where the exchange coefficients are defined with two terms, one representing turbulent transfer within the turbulent surface layer and one representing molecular transfer within the viscous sublayer following McPhee et al. (1987) (their Equation 11):."

Why the form with power '-1' is chosen if that power can be removed from this equation. In addition, I have not found this or similar expression in McPhee et al (1987), please clarify.

140 *The power of -1 was not necessary in these equations and has been removed. This form (without the -1 power) can be found in Equation 11 of McPhee et al. (1987) where the second term is the molecular contribution.*

Eq (14) $c_p$ should be removed, it makes the expression dimensionally inconsistent.

*Change has been made.*

L 146: 'It is noted by the PALM developers that this error was found to be small' –please provide a reference here.

145 *This note is embedded in the code itself. We provide the link here but we do not think it is an important reference for the readers as no further information is provided in this location in the code over that provided in the manuscript.* https://github.com/lanl/palm_lanl/blob/7f1444073b20ae81b451d4ab469d13e66b5358b7/trunk/SOURCE/surface_layer_fluxes_mod.f90#L2513-L2519

L 168: far field thermal driving is 0.15C, but in the figures elsewhere, there is no such case, only 0.1C. I believe one of those

150 is a misprint. It was actually the legend of these figures that did not show the thermal driving with sufficient precision. We have increased the precision such that the 0.15C case is labeled appropriately in all figures.

L 147: 'This error would be small if the first ∼10 cm were largely a constant flux layer, as hypothesized by McPhee (1983).' – I do not understand this sentence. Clearly, if 10 cm were a constant flux layer, then the mentioned fluxes would be equal. What was hypothesized by McPhee?

155 *We have edited the text to be a bit more specific about what we mean. We hoped to convey that the surface layer is thought to deviate only slightly from a constant flux layer. We also changed the citation to McPhee (1981), which contains this argument in more detail (though the 1983 paper provides the extension to scalar fluxes).*

"This error would be small if the first ∼10 cm were nearly a constant flux layer; McPhee (1981) hypothesized that the sub-ice surface layer would be nearly but not exactly a constant flux layer."

160 L 205: 'thermal driving increases from 0.5 to 0.6C ': there is no 0.5C case in the figure.

*We have corrected this typo to refer to the 0.4 °C thermal driving case.*

[Figure]

**Figure 2.** Time evolution of (a,d) domain-averaged, resolved turbulence kinetic energy (TKE), (b,e) melt rate and (c,f) friction velocity for (a-c) thermal driving simulations and (d-f) variable slope simulations. The black curve represents the same simulation in all panels in this and subsequent figures. The analysis window, the last inertial period, is shaded.

Fig. 1: the caption does not match the figure: (a) is not TKE but friction velocity etc.

*We have replaced Figure 1 with the version with TKE in panels a,d, now Figure 2.*

L 347: critical gradient Richardson number for development of hydrodynamic instabilities, and further transition to turbulence, $Ri_g$=0.25, has been first obtained by Miles and Howard in 1961.

*Thanks for the tip. This reference has been added.*

**3   Community comments**

I believe sections 1 and 2 would benefit greatly from a figure showing a sketch of the simulated flow, and some relevant physical quantities and conditions, e.g. the angles alpha and beta, etc.

*We have added a schematic figure, now labeled Figure 1.*

Also, if I'm not mistaking, the numerical methods used by the LES solver are not discussed. They should at least be mentioned briefly.

*This is a good suggestion, particularly as PALM has several methods available for advection, timestepping, and pressure. We have added the following text:*

[revised manuscript text omitted]